# Bird populations most exposed to climate change are less sensitive to climatic variation

Liam D. Bailey [1,2✉], Martijn van de Pol [1,3], Frank Adriaensen[4], Aneta Arct[5], Emilio Barba [6], Paul E. Bellamy [7], Suzanne Bonamour[8], Jean-Charles Bouvier[9], Malcolm D. Burgess [7,10], Anne Charmantier [11], Camillo Cusimano[12], Blandine Doligez[13], Szymon M. Drobniak [14,15], Anna Dubiec [16], Marcel Eens [17], Tapio Eeva [18,19], Peter N. Ferns [20], Anne E. Goodenough [21], Ian R. Hartley[22], Shelley A. Hinsley[23], Elena Ivankina [24], Rimvydas Juškaitis [25], Bart Kempenaers [26], Anvar B. Kerimov [27], Claire Lavigne [9], Agu Leivits [28], Mark C. Mainwaring[22], Erik Matthysen[4], Jan-Åke Nilsson [29], Markku Orell[30], Seppo Rytkönen [30], Juan Carlos Senar [31], Ben C. Sheldon [32], Alberto Sorace[33], Martyn J. Stenning[34], János Török [35], Kees van Oers [1], Emma Vatka [36], Stefan J. G. Vriend [37] & Marcel E. Visser [1]

The phenology of many species shows strong sensitivity to climate change; however, with few large scale intra-specific studies it is unclear how such sensitivity varies over a species' range. We document large intra-specific variation in phenological sensitivity to temperature using laying date information from 67 populations of two co-familial European songbirds, the great tit (*Parus major*) and blue tit (*Cyanistes caeruleus*), covering a large part of their breeding range. Populations inhabiting deciduous habitats showed stronger phenological sensitivity than those in evergreen and mixed habitats. However, populations with higher sensitivity tended to have experienced less rapid change in climate over the past decades, such that populations with high phenological sensitivity will not necessarily exhibit the strongest phenological advancement. Our results show that to effectively assess the impact of climate change on phenology across a species' range it will be necessary to account for intra-specific variation in phenological sensitivity, climate change exposure, and the ecological characteristics of a population.

---

A full list of author affiliations appears at the end of the paper.

Environmental temperature is often an effective predictor of future conditions that impact organismal fitness[1]. Because of this, many organisms exhibit a strong relationship between temperature and phenology, leading to clear phenological advancement with anthropogenic climate change[2–5]. The rate of phenological advancement is the product of a species' 'phenological sensitivity'[4] and 'climate change exposure' (Fig. 1), which are both affected by the biotic and abiotic environment[6–11]. For example, differences in the timing and availability of resources, due to factors such as habitat type, can affect phenology directly and also alter phenological sensitivity[6,10,11]. Similarly, climate change exposure can vary geographically, such as through Arctic amplification where the rate of climate change is greater at higher latitudes[7]. While it is well recognised that phenological sensitivity and climate change exposure can vary at an inter-specific level[4,5,12], it is poorly understood the extent to which these variables may vary, and covary, at an intra-specific level, and what consequences this may have for predicting 'phenological advancement' (Fig. 1) of a species over its entire range.

Estimation of a species' phenological sensitivity often relies on data from a small number of long-term wild study populations, or experimental studies, that are assumed to be representative of the species across its range. Yet there is evidence that populations can differ in their phenological sensitivity[13] and experimental studies may not reflect responses of wild organisms[14]. If there is substantial intra-specific variation in phenological sensitivity our ability to extrapolate from one or a few populations over a species' range will be limited. Variation in phenological sensitivity may be a function of underlying variation in biotic or abiotic variables, in which case results from one population may be unrepresentative of populations that experience different environmental conditions. Intra-specific variation could also be a consequence of genetic differences over a species range[15], so results from one population will be less representative of populations further away or separated by dispersal barriers[16]. The implications of intra-specific variation in phenological sensitivity

become more complex when considering the possibility of phenological mismatch[17,18], where phenological sensitivity of multiple species is relevant (e.g. food and prey). If patterns of intra-specific variation differ with trophic level this may lead to complex spatial patterns of trophic mismatch and organismal fitness[18]. Quantifying the extent of intra-specific variation in phenological sensitivity and identifying the drivers behind such variation, ideally in multiple species, represents a vital research topic if we hope to accurately predict the effects of future climate change.

Any attempt to understand intra-specific patterns of phenological sensitivity must first account for potential intra-specific differences in 'temperature windows'[19,20], the period during which temperature most strongly affects population phenology (Fig. 1). A population's temperature window will directly determine the temperature values used to quantify phenological sensitivity and climate change exposure, yet in many populations we have little a priori knowledge on the temperature window used by the organism and/or population under study[4,21,22]. Quantifying phenological sensitivity using incorrect temperature windows can lead to an underestimation of sensitivity[21]. Moreover, if appropriate temperature windows are used for some populations but not others, we may detect spurious intra-specific variation[23]. Similarly, using inappropriate temperature windows to calculate climate change exposure can lead to unreliable results as the effects of climate change may be stronger in some parts of the year than others[8,9]. Standardised methods to quantify temperature windows should be employed across populations to allow for more reliable estimation of phenological sensitivity and climate change exposure. This is best achieved by (re)analysis of original datasets from different studies, rather than by collating published results that often use a variety of different methods.

We use laying date information from two co-familial bird species, the great tit (*Parus major*) and blue tit (*Cyanistes caeruleus*), to study intra-specific variation in phenological sensitivity and climate change exposure. The great and blue tit are two of the most well

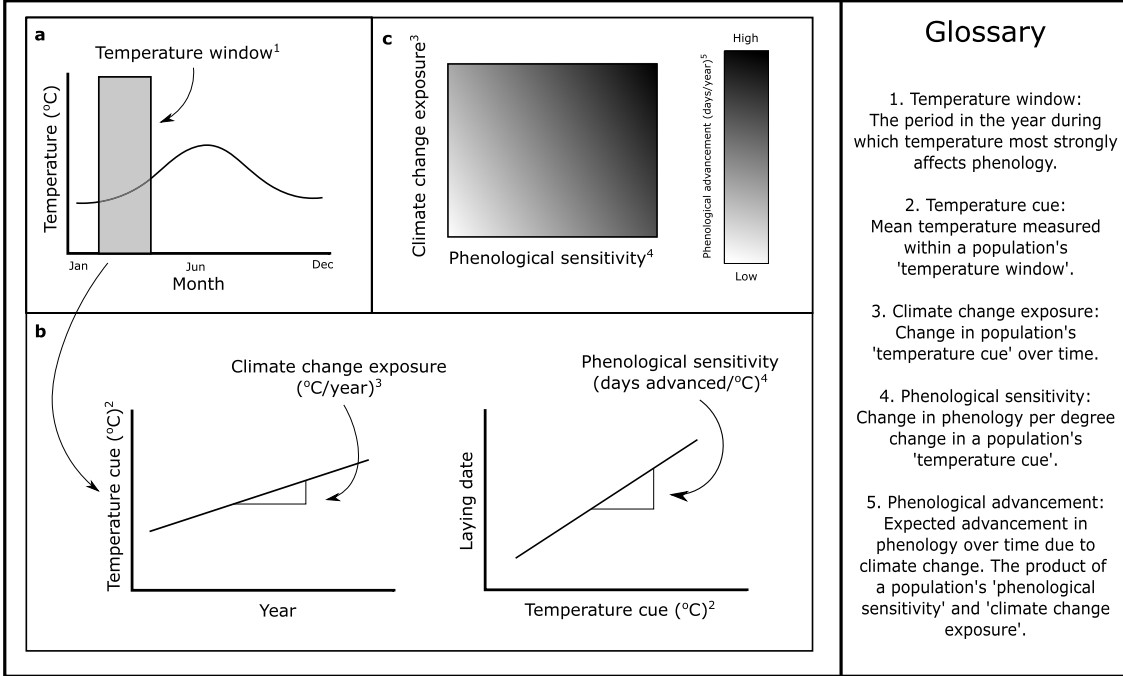

**Fig. 1 Illustration of key terms. a** Temperature window of a population (grey shaded area) is identified. **b** Population-specific temperature cue is used to estimate climate change exposure and phenological sensitivity. **c** Expected phenological advancement is the product of a population's climate change exposure and phenological sensitivity. Darker background colour represents higher phenological advancement.

studied bird species globally, with long-term nest-box breeding populations studied over a broad latitudinal and longitudinal range and throughout a diverse array of ecosystems[24]. These two species also provide a well-documented example of phenological advancement[13,25] and potential trophic mismatch as a result of climate change[18,26–28]. The number and diversity of great and blue tit populations and our detailed understanding of their phenology makes these species well suited for the goals of this paper. Furthermore, recent efforts at centralisation and standardisation of hole-nesting bird data allows analysis of great and blue tit data to be carried out easily and efficiently[24].

We used a standardised method to determine population-specific temperature windows for 67 populations of great and blue tits at a continental scale. We use these identified temperature windows to quantify population-specific phenological sensitivity and climate change exposure, and then test the impact of potential biotic (habitat) and abiotic (precipitation) variables on phenological sensitivity. Finally, we quantify (co)variance between phenological sensitivity and climate change exposure and estimate how intra-specific variation in these variables will affect phenological advancement. Climate change exposure is expected to show intra-specific variation with increasing exposure at higher latitudes[7]; however, patterns of intra-specific variation in phenological sensitivity are less well established. There is some evidence from Mediterranean evergreen habitats that phenological sensitivity is lower in these populations than those in deciduous forests[11]. Yet it is unclear whether these results are broadly generalisable in all evergreen ecosystems (e.g. needle-leafed, broad-leafed) and whether habitat type is still relevant at a continental scale, where other drivers may play a more dominant role. If habitat does drive phenological sensitivity at a continental scale then we would expect latitudinal trends in sensitivity due to the greater prevalence of evergreen habitats at higher latitudes, which would lead to possible covariance with latitudinal patterns in exposure.

We find intra-specific variation in phenological sensitivity in both the great tit and blue tit, mediated in part by the habitat characteristics in which a population resides. However, populations with higher sensitivity tend to have experienced less climate change exposure over the past decades, demonstrating that populations with high phenological sensitivity are not necessarily those that will exhibit the strongest phenological advancement.

By focussing on two widely studied species, this study has the potential to identify generalisable continental-scale drivers of intra-specific (co)variance in sensitivity and exposure that may also be relevant for rare or under-studied species in which long-term multi-population data are unavailable.

## Results

**Key results**. We documented strong intra-specific variation in phenological sensitivity, with a fourfold difference in phenological sensitivity across our study populations. Habitat type (deciduous, evergreen, mixed) was a key driver of this variation, with populations in deciduous habitats showing significantly stronger phenological sensitivity than those in either evergreen or mixed habitats. Due to contrasting latitudinal patterns in sensitivity and climate change exposure we observed a negative covariance between these two variables (Pearson's $r$: −0.56). Covariance between sensitivity and exposure meant that neither population characteristic alone was a particularly good correlate of phenological advancement (sensitivity $r$: 0.37; exposure $r$: 0.17). We present results from each section of the analysis in more detail below.

**Temperature windows**. We identified temperature windows in 70% of our populations (24/34 great tit, 23/33 blue tit). Temperature windows were detected more frequently in populations with more years of data (Supplementary Fig. 2), suggesting that failure to detect a temperature window after randomisation is due to limited sample size rather than providing evidence of phenological insensitivity. In all 47 populations where we detected a temperature window, we found that both blue tits and great tits had earlier laying date with warmer temperatures (Supplementary Data 1). None of the models showed evidence for significantly different responses between great and blue tits (Supplementary Tables 1, 4 and 5).

**Intra-specific variation in temperature windows**. There was a clear latitudinal pattern in temperature windows of both species, with populations at higher latitudes having temperature windows with midpoints later in the year ($\beta = 1.70$ days/degree latitude; 95% CI: 1.34/2.06; Fig. 2a; Supplementary Table 2). Temperature windows were also later in populations inhabiting evergreen

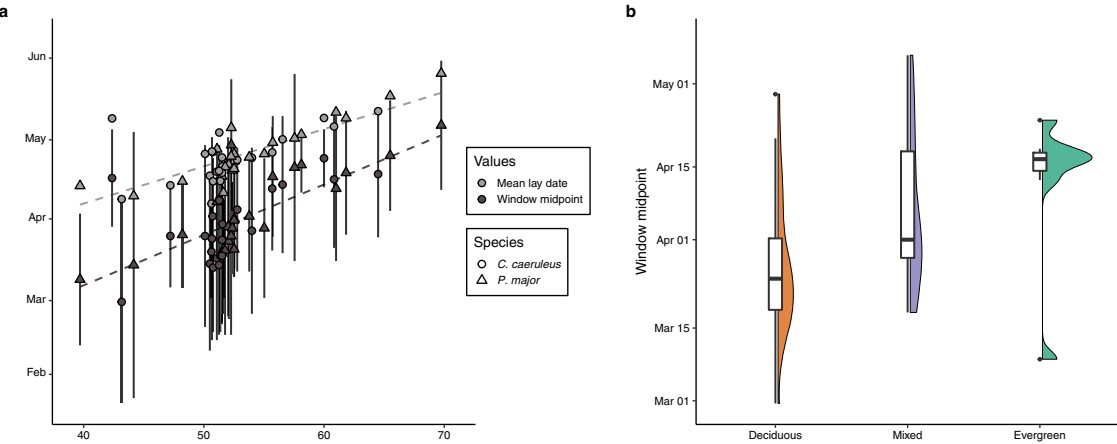

**Fig. 2 Intra-specific variation in temperature windows. a** Latitudinal change in temperature window midpoint (black) and mean annual laying date (grey) of European populations of great tits (triangles) and blue tits (circles). Temperature window midpoint increased with latitude in both species, while window duration (vertical lines) and the delay (difference between window midpoint and mean laying date; dashed lines) did not change significantly. **b** Difference in temperature window midpoint for great and blue tits inhabiting deciduous (orange), mixed (blue), or evergreen (green) habitats ($n = 27$, 13, and 7 biologically independent populations respectively). Shown are box and violin plots of temperature window midpoints of both great and blue tits in each habitat type. Boxplots shows median (centre line), 25th and 75th quantiles (lower and upper hinges), and 1.5× inter-quartile range (whiskers). Observations further than 1.5× inter-quartile range are shown as outlier points.

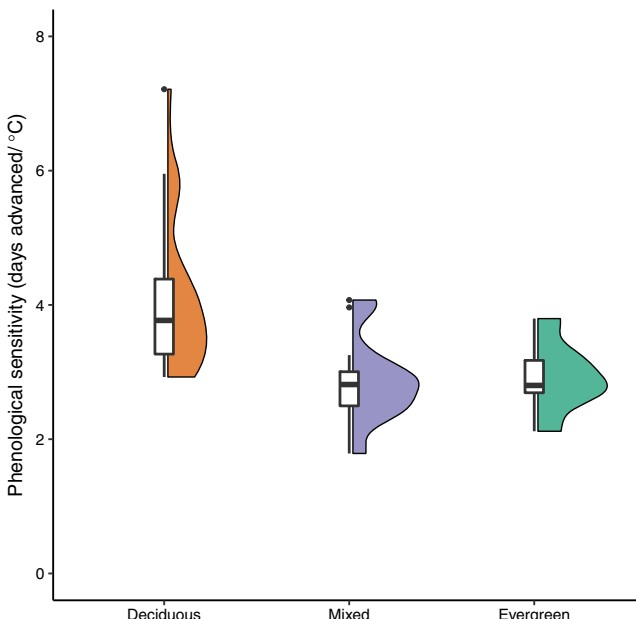

**Fig. 3 Variation in phenological sensitivity.** Phenological sensitivity (days advanced/C) differed between habitat types: deciduous (orange), mixed (blue), or evergreen (green; n = 27, 13, and 7 biologically independent populations respectively). Phenological sensitivity is higher in deciduous habitats than either evergreen or mixed habitats. Shown are box and violin plots of phenological sensitivity of both great and blue tits in each habitat type. Boxplots shows median (centre line), 25th and 75th quantiles (lower and upper hinges), and 1.5× inter-quartile range (whiskers). Observations further than 1.5× inter-quartile range are shown as outlier points.

forests compared to those in deciduous forests ($\beta = 9.56$ days; 95% CI: 4.86/14.56; Fig. 2b; Supplementary Table 2). Temperature window midpoint showed no clear relationship with longitude, nor was there any relationship between the temperature window duration and the latitude or longitude of populations in either species (Supplementary Fig. 3; Supplementary Table 2). The average delay between temperature window midpoint and mean laying date were 26.5 (Range: 8.59–41.01). Temperature window delay did not change significantly with latitude, but there was some evidence of shorter delays in more eastern populations (Supplementary Fig. 3; Supplementary Table 3).

**Drivers of phenological sensitivity.** We observed a fourfold difference in the phenological sensitivity, ranging from 1.8 days/°C (Estonia) to 7.2 days/°C (Okehampton, UK; Supplementary Data 1). Sensitivity was significantly associated with habitat type (deciduous, mixed, or evergreen) in both species (Fig. 3), with significantly higher sensitivity in deciduous habitats than either evergreen or mixed habitats ($\beta_{\text{EVERGREEN}} = -0.86$ days/°C; 95% CI: $-1.28/-0.44$; $\beta_{\text{MIXED}} = -0.74$ days/°C; 95% CI: $-1.05/-0.37$; Supplementary Table 3). Phenological sensitivity was also higher in populations with higher annual precipitation (Supplementary Table 3). Great tit populations were significantly more sensitive than blue tits, although the magnitude of this difference was much less than that of habitat or precipitation (Supplementary Table 3). We observed no pairwise correlation between populations more than 5° apart (see Supplementary Methods for more details on spatial autocorrelation).

**Covariance between sensitivity and climate change exposure.** Climate change exposure over the past seven decades varied from 0.01 °C/year (East Dartmoor, UK) to 0.05 °C/year (Upeglynis,

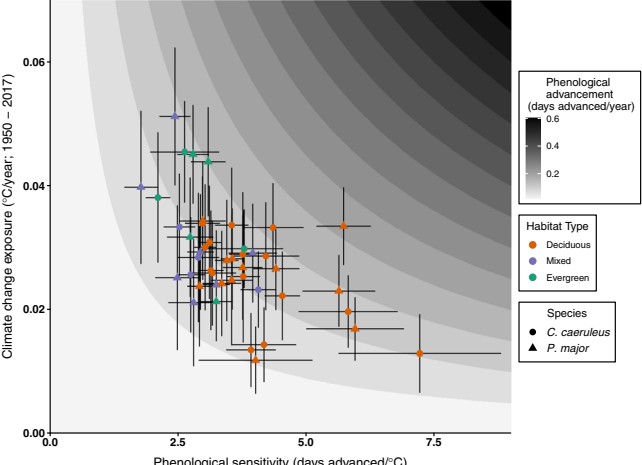

**Fig. 4 Estimated phenological advancement.** Phenological advancement (days advanced/year of great tit (*Parus major*; triangle) and blue tit (*Cyanistes caeruleus*; circle) populations in deciduous (orange), mixed (blue), and evergreen (green) dominant habitats. Phenological advancement (background colour) is a product of phenological sensitivity and climate change exposure. A darker background colour represents greater phenological advancement over time. Populations with the highest recorded phenological sensitivity do not show the highest expected phenological advancement due to their lower climate change exposure. Coloured points represent each population's expected phenological advancement, the product of estimated phenological sensitivity and climate change exposure. Note that phenological sensitivity and climate change exposure are calculated within population-specific temperature windows, which accounts for observed intra-specific variation in temperature window midpoints (Fig. 2). Vertical and horizontal lines represent standard errors of slope estimates for climate change exposure and phenological sensitivity respectively. Number of biologically independent years used to estimate phenological sensitivity for each population is available in Supplementary Data 1. Climate change exposure is estimated in each population using 68 biologically independent years (1950–2017).

Lithuania). Those populations with the highest phenological sensitivity tended to have experienced lower climate change exposure (Fig. 4; Supplementary Data 1), with a negative correlation observed between phenological sensitivity and climate change exposure following non-parametric bootstrapping with 5000 iterations (Pearson's $r$: −0.56; Supplementary Fig. 4). The covariance result cannot be explained by any mathematical dependency between the two variables (Supplementary Methods). Expected phenological advancement, the product of sensitivity and exposure, showed a fourfold difference among populations (0.05–0.19 days advanced/year), a similar magnitude of variation to that seen in phenological sensitivity and climate change exposure. There was a low correlation between phenological advancement and both sensitivity and exposure (sensitivity $r$: 0.37; exposure $r$: 0.17). Great and blue tits in Gotland (Sweden) had the largest expected advancement, while those in East Dartmoor (UK) had the smallest.

## Discussion
We studied phenology in 67 populations of great and blue tits over a large part of their breeding range, within 47 of which we were able to identify temperature windows. As expected, earlier laying dates coincided with warmer conditions in all 47 populations (Supplementary Data 1). While all populations laid earlier at higher temperatures, there was a fourfold difference in the strength of phenological sensitivity. Variation in phenological sensitivity was associated with ecological characteristics, with stronger sensitivity observed in deciduous dominated habitats

and areas with higher annual precipitation. We also document large (fourfold) intra-specific variation in climate change exposure; however, due to contrasting latitudinal patterns in habitat type and climate change exposure we ultimately observed a negative covariance between exposure and sensitivity across our study populations. While populations of great and blue tits inhabiting deciduous habitats showed strong sensitivity, many of these populations are found at mid-latitudes (e.g. UK, Netherlands) and so have lower climate change exposure. In contrast, more northerly populations have experienced high climate change exposure over the past decades[7], yet many of these populations reside in evergreen habitats and so have weaker phenological sensitivity.

Large intra-specific variation in phenological sensitivity and climate change exposure shows that phenological advancement cannot be confidently predicted using either of these variables alone, but rather requires an estimation of both variables. The combined importance of sensitivity and exposure has been discussed extensively in an inter-specific context[29–31], but we show here that intra-specific (co)variance in these variables is also relevant. The presence of such large intra-specific variation highlights the limitations of using single populations to draw conclusions about the impacts of climate change across a species' range. Sampling from multiple populations with diverse ecological characteristics over a broad geographic scale will likely provide a more reliable estimate of climate change impacts. In rare or poorly studied species, where such long-term multi-population data is unavailable or infeasible, accounting for potential drivers of intra-specific variation wherever possible will be important to better quantify impacts of climate change.

One major driver of phenological sensitivity in great and blue tits was habitat type (Fig. 3). Our results show that habitat type can not only impact phenology directly[32–34] but can also affect phenological sensitivity, confirming results from previous analyses conducted at a smaller spatial scale[11]. Habitat type effects were apparent over a broad latitudinal range, with low phenological sensitivity observed in both Fennoscandian needle-leafed evergreen habitats (e.g. Askainen, Finland) as well as Mediterranean broad-leafed evergreen habitats (e.g. Corsica, France; Supplementary Data 1). The fact that this pattern was observed in both needle-leafed and broad-leafed evergreen forests suggests that low phenological sensitivity is not restricted to a single floral species assemblage but is a common trait of evergreen habitats.

Differences in food resources between habitat types may help explain the observed sensitivity patterns. Temperature is a laying date cue in great and blue tits[25], enabling breeding birds to synchronise peak offspring provisioning with the peak in caterpillar abundance. Evergreen habitats tend to have lower peak caterpillar abundance during the breeding season than deciduous systems[34,35], which may necessitate greater dietary flexibility in nestlings and reduce the reliance of breeding birds on caterpillars and corresponding temperature cues[36]. Dietary differences between habitat types have been documented in Mediterranean great and blue tits[35,37], and this pattern may apply more broadly across Europe. Evergreen forests also tend to have a wider period of peak abundance than deciduous systems[34], which may affect the selection landscape. A narrow resource peak in deciduous habitats will lead to a high fitness cost of phenological mismatch[18,26–28], creating strong selective pressure for synchronisation by tracking temperature cues. In comparison, a broad resource peak in evergreen habitats will reduce the costs of asynchrony, leaving birds less strongly temperature constrained. Finally, temperature cues have been shown to provide a less reliable indicator of resource abundance in evergreen than deciduous habitats[11], which may lead populations to rely on alternative cues, such as vegetation phenology[33].

If observed habitat patterns are a consequence of resource availability, we predict that other insectivorous passerines that rely on peaks in invertebrate abundance for offspring provisioning will show similar differences in phenological sensitivity between habitat types, although this will likely depend on the species' dietary specialisation. These patterns may also extend beyond this specific feeding guild. For example, a relationship between patterns of resource availability and phenology has also been proposed for breeding shorebirds in Greenland[10], suggesting that the importance of food peak structure on phenology may be broadly generalisable to any species that relies on a seasonal peak in resources and is therefore vulnerable to phenological mismatch. Some examples may include invertebrates reliant on budding vegetation[38], secondary and tertiary consumers that utilise peaks in juvenile prey[38] or migratory arrivals[39], and plants reliant on insect pollinators[40]. If habitats differ in the abundance, predictability, or the length of availability in seasonal resources this may drive intra-specific variation in phenological sensitivity. In fact, differences in phenological sensitivity between our two sympatric study species was much smaller than the differences due to ecological characteristics. This suggests that drivers such as habitat type may be better predictors of sensitivity than species traits, as has also been observed in body condition responses to climate change[41]. In general, we predict that species that show dietary specialisation and occupy habitats with narrow resource peaks will show stronger phenological sensitivity.

As we analysed data at the population level, observed patterns could reflect both individual-level responses to temperature (i.e. phenotypic plasticity) and structuring in the data[42]. If late-nesting birds forgo reproduction in warmer years we would still detect a relationship between laying date and temperature without any phenotypic plasticity[42]. Even if phenotypic plasticity explains observed differences in sensitivity, we cannot be sure that our results reflect intra-specific differences in the selective landscape or are due to differences in the optimality of individual responses[43]. Previous work in a subset of our study populations has demonstrated that differences in phenological sensitivity between populations can be attributed to individual-level differences[11] and that populations tend to show similarly optimal levels of phenological change even as the phenological optimum shifts across latitudes[44]. With these previous results in mind, we assume that population-level phenological sensitivity in our analysis reflects optimal shifts in individual phenology.

Quantification of phenological sensitivity and climate change exposure in our analysis relies on temperature data from the European wide E-OBS Gridded Dataset. Gridded datasets are an invaluable tool to study macroclimatic temperature change at a broad spatiotemporal scale, but such datasets do not account for local microclimatic variation that is more relevant to organismal behaviour[45,46]. Gridded datasets can differ from local microclimatic temperature by as much as 4 °C[46], mediated by both habitat and landscape scale characteristics, such as canopy cover and degree of habitat fragmentation[46,47]. Although macro- and microclimatic trends in climate change exposure are strongly related, failure to account for microclimates will likely contribute to additional unexplained variation in phenology[48], which may be more pronounced in those habitats where macro- and microclimatic temperatures diverge most, such as those near urban areas[46]. Detailed fine-scale temperature measurements were unavailable over the large spatiotemporal scale used in our study, making it impossible to directly incorporate effects of microclimatic variation in our analysis. In the future, accounting for differences in habitat characteristics that are known to influence microclimate may provide a feasible alternative to directly measuring microclimates particularly for variables that can be estimated through remote sensing, such as canopy cover or habitat fragmentation[49,50].

Although our results show clear intra-specific variation in phenological sensitivity, this does not provide an effective predictor of a population's phenological advancement. To properly understand phenological advancement, we must account for (co) variance in both phenological sensitivity and climate change exposure. We observed a clear positive latitudinal pattern in climate change exposure, with more northerly populations experiencing higher exposure, following the well-documented pattern of Arctic amplification[7]. In contrast, the greater prevalence of (needle-leafed) evergreen habitats at northern latitudes meant that those northerly populations with higher climate change exposure tended to be less sensitive. Ultimately, this negative covariance between sensitivity and exposure meant that phenological sensitivity and climate change exposure of a population were both poor predictors of the considerable intra-specific variation in phenological advancement in our study species (Supplementary Methods).

The negative covariance pattern we observed between sensitivity and exposure appears to be a consequence of latitudinal patterns in habitat type. Whether negative covariance between sensitivity and exposure will occur in other species, or when studying other traits, will therefore depend on the underlying ecological drivers of trait variation and how such drivers vary over a species' range. Furthermore, we should not assume that observed covariance patterns will remain stable over time if populations experience shifts in the biotic or abiotic environment. For example, phenological sensitivity of a population would be expected to change if the dominant species in a habitat changes over time (e.g. evergreen to deciduous dominant), as might be expected due to species range shifts with future climate change. Therefore, our ability to predict impacts of climate change in populations will depend both on our ability to quantify covariance between sensitivity and exposure and also our ability to predict future changes in those environmental drivers that affect sensitivity.

The fitness consequences of expected phenological advancement that we report here can only be understood relative to phenological advancement in other selective agents, such as food resources, that provide a phenological 'yardstick'[51]. Differences in phenological advancement between breeding birds and their food resources will lead to increasing phenological mismatch and negative fitness consequences over time[18]. Such differences may occur due to corresponding differences in phenological sensitivity as well as differences in temperature windows and corresponding temperature cues used by predator and prey species[17]. We expect that intra-specific variation in phenological advancement should also occur in the primary food species of great and blue tits, yet we currently have no understanding of the direction or magnitude of such variation nor whether such variation will be driven by the same biotic and abiotic variables. This issue is further complicated by the fact that the dominant prey species of a population can also differ[36], such that variation in the phenological 'yardstick' of our study species will be caused by both intra- and inter-specific variation in different prey species. We currently lack corresponding continental-scale data on the phenological sensitivity and climate change exposure of the bird's food resources, and we cannot be sure how our observed (co) variance patterns between sensitivity and exposure may reflect intra-specific variation in phenological asynchrony, and fitness consequences, across Europe[17,52]. Our results therefore represent a first step towards a broader understanding of phenology and trophic interactions at a continental scale.

## Methods

**Study populations**. We used the SPI-Birds Network and Database[24] to collate data from 67 populations of two closely related insectivorous passerines, the great tit (*Parus major*) and blue tit (*Cyanistes caeruleus*), across Europe (34 great tit and 33 blue tit; in 27 cases data for both species were collected from the same study site Fig. 5). Both species nest in tree cavities but will also use artificial nest-boxes where provided. All data used in this study are derived from nest-box breeding populations.

We limited our analyses to populations for which a minimum of 9 years of data were available as we have previously been able to quantify temperature windows in a dataset of the same length[53]. Sampled populations ranged latitudinally from 37.6° N (Italy) to 69.8° N (Finland), with the northern most populations close to the northern range limit of both species[19,54]. Populations ranged in longitude from −3.99° W (UK) to 36.85° E (Russia). Populations were

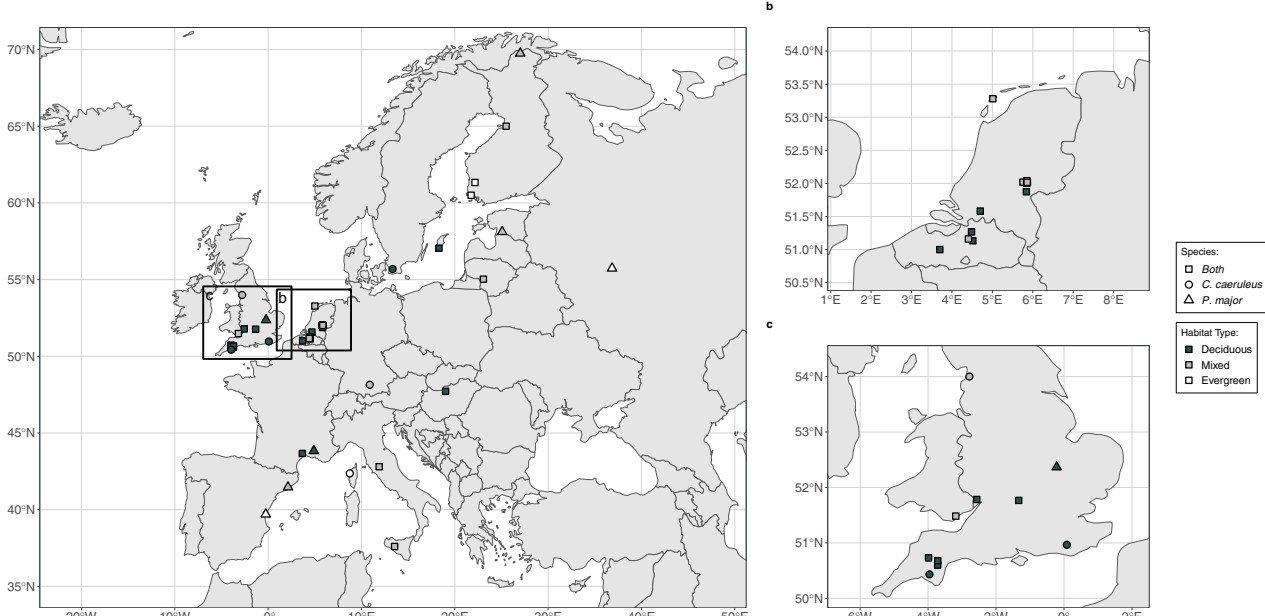

**Fig. 5 Distribution of study populations from which phenological data were collected. a** Phenological information (laying date) was recorded at a site for either great tit (*Parus major*; triangle), blue tits (*Cyanistes caeruleus*; circle), or both species (square). Data was collected on both species at the majority of sites (68%; 27 sites). Sites are classified as either deciduous dominant (black), evergreen dominant (white), or mixed (grey). Insets (**b**, **c**) show populations across Belgium and Netherlands and the UK respectively.

sampled from a range of habitats dominated by either deciduous or evergreen tree species or a mix of both. The habitat type of each population was defined using site descriptions from data owners. Populations with deciduous dominant tree species (*Quercus petraea, Q. pubescens, Q. robur, Q. cerris, Q. cerrioides, Fagus spp., Betula spp., Fraxinus excelsior, Acer campestre, A. pseudoplatanus*) were defined as 'deciduous' ($n = 33$), while those with evergreen dominant species (*Quercus ilex, Pinus spp., Picea spp.*) were defined as 'evergreen' ($n = 7$). Populations with both deciduous and evergreen dominant species were considered mixed ($n = 27$). Two populations (Sagunto, Spain; Avignon, France) were situated in plantations of orange (*Citrus × aurantium*; evergreen) and apple/pear (*Malus domestica* and *Pyrus spp.*; deciduous) respectively. Due to limited sample size, populations in broad-leafed ($n = 2$) and needle-leafed ($n = 5$) evergreen habitats were grouped together.

**Phenological data**. We quantified laying date (the date on which the first egg of a clutch was laid) for all females in all populations based on regular nest-box checks (at least weekly). When nests were not observed on the day the first egg was laid, laying date was estimated assuming one egg laid per day. For all analyses we used the laying date of first clutches and excluded second and replacement clutches. First clutches were defined as those laid within 30 days of the earliest clutch of the season in a given year and population[55]. We used the mean laying date of first clutches within a year as a measure of population phenology for all further analyses.

**Climate data**. We obtained mean daily temperature data (°C) from the E-OBS Gridded Dataset v17.0 with a resolution of 0.25 degrees[56]. The gridded dataset uses blended weather time series from the European wide weather station network of the European Climate Assessment & Dataset project (ECA&D; https://www.ecad.eu/). Blended time series use data from nearby weather and synoptic stations to extend and fill in gaps in existing weather station time series. The E-OBS Gridded Dataset uses all observations from available blended time series that are considered 'valid' under the ECA&D quality control rules[57]. Full documentation explaining blending and quality control methods can be found on the ECA&D website (https://www.ecad.eu/).

For every population, we extracted daily mean temperature (°C) for all years in which phenological data were available. In six populations, the study site location did not overlap with the gridded dataset. In four of these cases (Sagunto, Spain; Barcelona, Spain; Cardiff, UK; Askainen, Finland), we extracted temperature data from the nearest grid cell instead. The alternative grid cells were never more than 8 km from the study site (3–8 km). In one case (Vlieland, Netherlands) we interpolated daily mean temperature information from weather stations provided by the Royal Dutch Meteorological Institute (KNMI). All weather stations used to interpolate Vlieland data were from the neighbouring island of Terschelling (<33 km away). As Vlieland is an island population, we considered local weather station data from a neighbouring island to be more reliable than the nearest grid cell located on the Dutch mainland. In the final case (Sicily, Italy), temperature data were taken from weather stations operated at the study site by a co-author (C. Cusimano). To ensure that our results were robust to the choice of temperature data in Vlieland and Sicily we conducted supplementary analyses with these populations excluded (Supplementary Methods). All results were consistent with these populations removed.

**Estimating temperature windows**. There are a range of methods available to determine population temperature windows[23,58]. We employed an absolute sliding time window approach[58] using the R package climwin[21,22] and following the workflow described by van de Pol et al.[22] (their Fig. 3). In brief, we first defined an intercept only 'baseline model'. We next defined our climatic variable of interest and decided on the attributes of our sliding window approach, including the maximum and minimum range of possible windows and the aggregate statistic (here the mean) applied to each temperature window. Finally, we ran our sliding window analysis and employed a randomisation approach to account for false positives due to multiple testing. Each of these steps is described in more detail below.

For our baseline model, we used a general linear model with a Gaussian error distribution. Model residuals were weighted by the inverse of the standard error in annual mean laying date to account for uncertainty in this value.

Following previous studies on these species, our analysis focussed on mean temperature[4,20,59]. We used an absolute sliding window approach (i.e. we assume that all individuals in a population had the same temperature window), and tested for all potential temperature windows over a 365-day period before June 1st. Duration of temperature windows was allowed to vary between 1 and 365 days. We determined the mean temperature within each potential temperature window and estimated the relationship between this temperature and mean annual laying date. Following previous studies, we tested for linear relationships between temperature and laying date[4].

As we tested a large number of potential temperature windows there were inherent risks associated with multiple testing[22]. To address this issue, we randomised the order of the original data in each population to remove any relationship between

temperature and laying date and then re-ran the sliding window analysis. We replicated this randomisation procedure 100 times. We then compared our observed result to that of our 100 randomisations and determined the probability that our observed result could occur in a dataset where no relationship exists between temperature and laying date. This method is described in detail by van de Pol et al.[22]. We used the metric $P_{\Delta AICc}$ to assess the probability that the identified temperature window was a false positive[21,22]. $P_{\Delta AICc}$ represents the probability that we would observe a given $\Delta AICc$ value in our temperature window analysis when no relationship exists between temperature and phenology[22].

We considered a temperature window to represent a true temperature cue if $P_{\Delta AICc}$ was ≤0.05 (i.e. the chance of such a result occurring in a randomised dataset was ≤5%). Populations with a best temperature window ≤14 days in duration were also excluded, as such short windows are biologically less plausible and can produce statistical artefacts[22]. Populations without a true temperature cue were not used for further analyses.

In all populations where we identified a true temperature cue, we determined the duration and midpoint of the temperature window and the difference between the midpoint and the mean laying date of the population over the study period (delay). The window midpoint is a useful single metric to describe the position of the temperature window within a calendar year and is expected to be later for more northerly populations[59] and those in evergreen habitats due to later resource peaks[34]. Temperature window delay represents the time between when a cue is detected and a response (i.e. egg laying) is exhibited. Delay between a cue and response may be caused by physical restrictions that limit the speed of response (e.g. gonadal development[16]) or may reflect a delay between cues and relevant future conditions when cues act through multiple trophic levels[44]. Previous studies have assumed that intra-specific variation does not exist in this delay variable, but this has not been explicitly tested[13]. Variation in temperature window duration with latitude has been reported in previous studies[20,59,60]; however, the direction of such an effect has differed between studies making any expectations around duration unclear.

**Phenological sensitivity to temperature**. Our sliding window method estimated the relationship between mean temperature and laying date in the best supported temperature window. However, a correlation between temperature and laying date may arise due to unmeasured, non-climatic variables that lead to a shared temporal trend between temperature and laying date[61]. We used structural equation models to quantify the relationship between temperature and laying date after accounting for shared trends over time using the R package lavaan[62], using temperature data from population-specific temperature windows. We partitioned the effect of time (year) on laying date into a direct and indirect pathway. The indirect pathway accounted for effects of time that occur via changes in temperature, such as climate change. The direct pathway accounted for any changes in laying date over time that occurred through effects other than temperature change. The relationship between temperature and laying date in the structural equation model was used as the measure of phenological sensitivity. A path diagram representing the structural equation model is included as Supplementary Fig. 1. We used non-parametric bootstrapping with 1000 iterations to estimate standard errors of coefficients in the structural equation models.

**Intra-specific variation in temperature windows**. We first analysed intra-specific variation in temperature window characteristics. To understand variation in the midpoint, duration, and delay of temperature windows we built general linear mixed effects models with a Gaussian error term. Population ID was included as a random intercept to account for cases where both species were sampled at the same site (Fig. 5). Population intercepts were assumed to be normally distributed with mean = 0 and variance = $\sigma^2$. Our models included latitude, longitude, species, and habitat type (deciduous, evergreen, or mixed) as fixed effects, plus an interaction between species and latitude and longitude.

**Biotic and abiotic drivers of phenological sensitivity**. After accounting for variation in temperature windows we next assessed which variables best explain phenological sensitivity in different populations. We fitted a general linear mixed effects model with a Gaussian error term with population ID included as a random intercept. We included a fixed effect to account for differences in annual precipitation patterns between populations. We derived a precipitation metric from the principal component analysis of Metzger et al.[63] which incorporates variation in precipitation (mm) across multiple months over the year to provide a quantification of annual precipitation patterns at each study site. A higher PCA value represents a site with higher precipitation. We had no clear expectation for the period during which precipitation should affect phenology, therefore we included this broad measure of precipitation patterns. We also included an effect of habitat type (deciduous, mixed, or evergreen) and an interactions between species and each of our fixed effects. To account for potential spatial autocorrelation in phenological sensitivity we included a Matérn correlation function as a random intercept term, which estimates pairwise correlations between points as a function of their Euclidian distance, using the spaMM package in R[64].

**Intra-specific variation in phenological advancement.** We next calculated the expected phenological advancement of each population as the product of its phenological sensitivity and climate change exposure (Fig. 1). To ensure that climate change exposure was comparable across populations with different lengths and start dates, we calculated climate change exposure within a standard period (1950–2017) for all populations. 1950–2017 represents the full temporal range of the E-OBS Gridded Dataset v17.0. This decision will mean that climate change exposure and phenological sensitivity are estimated over different time periods for each population, and so we are quantifying *expected* phenological advancement, which represents the product of phenological sensitivity and climate change exposure assuming that phenological sensitivity of each population has remained unchanged between 1950 and 2017. Phenological advancement was only assessed in populations where a temperature window was identified and was not assessed for populations where no temperature data were available from the gridded dataset (Sicily and Vlieland).

**Additional statistical details.** For all general linear mixed effects models, we calculated 95% confidence intervals using parametric bootstrapping with 1000 iterations. Generalised variance inflation factors were used to test for multi-collinearity between variables in all models with a cut-off value of 3. No variables exceeded our variance inflation factor cut-off, suggesting that multi-collinearity between model predictors was not a concern. All analyses were conducted using R (v. 4.0.3)[65] in RStudio (v. 1.3.959).

**Reporting summary.** Further information on research design is available in the Nature Research Reporting Summary linked to this article.

## Data availability
The phenology data and population characteristics data used in this study and the temperature data used to run sliding window analysis for Sicily and Vlieland are available in the Zenodo repository (https://doi.org/10.5281/zenodo.5747635)[66]. The E-OBS Gridded Dataset v17.0 is freely available on request from the European Climate Assessment & Dataset project (ECA&D; https://www.ecad.eu/). The data generated from sliding time window analysis, randomisation and fitting of structural equation models are available in the Zenodo repository (https://doi.org/10.5281/zenodo.5747635)[66]. A summary of results for each population generated in this study is also provided in the Supplementary Data 1.

## Code availability
All code used for analyses is stored in the GitHub repository LiamDBailey/baileyetal2021 and archived on Zenodo, (https://doi.org/10.5281/zenodo.6027546)[67].

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

## Acknowledgements

We would like to give a special acknowledgement to all the fieldworkers who have been invaluable in helping collect these many decades of data. We acknowledge the E-OBS dataset from the EU-FP6 project UERRA (http://www.uerra.eu) and the Copernicus Climate Change Service, and the data providers in the ECA&D project (https://www.ecad.eu). Field study in Moscow region (E.I. and A.B.K.) was supported by Russian Science Foundation (RSF Grant No. 20-44-01005). This study was also funded by research project CGL-2020 PID2020-114907GB-C21 (to J.C.S.) from the Ministry of Economy and Competitivity, Spanish Research Council. A.C. was funded by the European Research Council (Starting grant ERC-2013-StG-337365-SHE). J.T. was funded by the Hungarian National Research, Development and Innovation Office (K-115970). S.M.D. was supported by the Discovery Early Career Fellowship (Australian Research Council, grant no. DE180100202). Long-term study on Gotland continues thanks to access provided by local landowners, and receives continuing funding from the Polish National Science Centre (most recently grants no. 2020/39/B/NZ8/01274 and 2015/18/E/NZ8/00505).

## Author contributions

L.D.B., M.v.d.P., M.E.V. conceived of the study. L.D.B. and S.J.G.V. located and compiled data across all study populations. F.A., A.A., E.B., P.E.B., S.B., J.C.B., M.D.B., A.C., C.C., B.D., S.M.D., A.D., M.E., T.E., P.N.F., A.E.G., I.R.H., S.A.H., E.I., R.J., B.K., A.B.K., C.L., A.L., M.C.M., E.M., J.Å.N., M.O., S.R., J.C.S., B.C.S., A.S., M.J.S., J.T., K.v.O., E.V. and M.E.V. were responsible for collecting and curating phenology data. L.D.B. analysed data, produced figures, and took the lead in writing the text. All authors discussed the results and contributed to writing the paper.

## Competing interests

The authors declare no competing interests.

## Additional information

[1]Department of Animal Ecology, Netherlands Institute of Ecology (NIOO-KNAW), Wageningen, The Netherlands. [2]Department of Evolutionary Genetics, Leibniz Institute for Zoo and Wildlife Research (IZW), Berlin, Germany. [3]College of Science and Engineering, James Cook University, Townsville, QLD, Australia. [4]Evolutionary Ecology Group, Department of Biology, Universiteitsplein 1, University of Antwerp, Antwerp, Belgium. [5]Institute of Systematics and Evolution of Animals, Polish Academy of Sciences, Kraków, Poland. [6]'Cavanilles' Institute of Biodiversity and Evolutionary Biology, University of Valencia, Valencia, Spain. [7]RSPB Centre for Conservation Science, The Lodge, Sandy, Bedfordshire, UK. [8]Sorbonne Université, Centre d'Écologie et des Sciences de la Conservation (UMR 7204), Muséum National d'Histoire Naturelle, Paris, France. [9]INRAE, PSH, Plantes et Systèmes de culture Horticoles, Avignon, France. [10]Centre for Research in Animal Behaviour, University of Exeter, Exeter, UK. [11]Centre d'Ecologie Fonctionnelle et Evolutive, CNRS, EPHE, IRD, Univ Montpellier, Montpellier, France. [12]Stazione Ornitologica Aegithalos, Palermo, Italy. [13]Laboratoire de Biométrie et Biologie Evolutive, CNRS UMR 5558, University of Lyon, Université Claude Bernard Lyon 1, Lyon, France. [14]Institute of Environmental Sciences, Jagiellonian University, Kraków, Poland. [15]Ecology & Evolution Research Centre; School of Biological, Environmental and Earth Sciences, University of New South Wales, Sydney, NSW, Australia. [16]Museum and Institute of Zoology, Polish Academy of Sciences, Warszawa, Poland. [17]Behavioural Ecology & Ecophysiology Group, Department of Biology, University of Antwerp, Antwerp, Belgium. [18]Department of Biology, University of Turku, Turku, Finland. [19]Kevo Subarctic Research Institute, University of Turku, Turku, Finland. [20]Cardiff School of Biosciences, Cardiff University, Cardiff, UK. [21]School of Natural and Social Sciences, Francis Close Hall, University of Gloucestershire, Cheltenham, UK. [22]Lancaster Environment Centre, Lancaster University, Lancaster, UK. [23]UK Centre for Ecology & Hydrology, Wallingford, UK. [24]Zvenigorod Biological Station, Lomonosov Moscow State University, Moscow, Russia. [25]Nature Research Centre, Vilnius, Lithuania. [26]Department of Behavioural Ecology and Evolutionary Genetics, Max Planck Institute for Ornithology, Seewiesen, Germany. [27]Department of Vertebrate Zoology, Faculty of Biology, Lomonosov Moscow State University, Moscow, Russia. [28]Department of Nature Conservation, Environmental Board, Tallinn, Estonia. [29]Evolutionary Ecology, Department of Biology, University of Lund, Lund, Sweden. [30]Department of Ecology and Genetics, University of Oulu, Oulu, Finland. [31]Evolutionary and Behavioural Ecology Research Unit, Museu de Ciències Naturals de Barcelona, Barcelona, Spain. [32]Edward Grey Institute, Department of Zoology, University of Oxford, Oxford, UK. [33]ISPRA, Rome, Italy. [34]School of Life Sciences, University of Sussex, Sussex, East Sussex, UK. [35]Behavioural Ecology Group, Department of Systematic Zoology and Ecology, ELTE Eötvös Loránd University, Budapest, Hungary. [36]Ecological Genetics Research Unit, Organismal and Evolutionary Biology Research Programme, Faculty of Biological & Environmental Sciences, University of Helsinki, Helsinki, Finland. [37]Centre for Biodiversity Dynamics, Department of Biology, Norwegian University of Science and Technology, Trondheim, Norway. ✉email: liam.bailey@liamdbailey.com

