## [Peer Review File · Nature Communications]

Reviewer comments -

Reviewer #1 (Remarks to the Author):

This study examines the relationship of date of first egg laid in two species, Great Tit and Blue Tit, and temperature windows across 67 nest-box breeding populations from a wide range of latitude and longitude across Europe. The paper finds that the temperature windows to which these species select laying dates are later at higher latitudes, but no relationship with longitude. Interestingly, the rate of phenological change does not positively scale with changing temperature, rather the opposite. The authors also explore three different broad habitat class, deciduous forest, mixed forest, and evergreen show differing levels of phenological sensitivity, with deciduous forests showing the highest sensitivity (days advanced/degree C). Mixed and evergreen habitats showed comparable rates of phenological sensitivity. The paper is well written and contains few errors throughout. I appreciate the authors taking the time to address comments from a previous review. Figure 1 now nicely describes and illustrates the terms used throughout the paper.

I have no further comments to improve this paper.

Reviewer #2 (Remarks to the Author):

The key results of this paper are to show that there is intra-specific spatial variation in phenological sensitivity and climate exposure; and, in particular that these two measures are negatively correlated in this case. This means that existing methods which use single populations to estimate climate sensitivity across a range, and/or where only one of phenological sensitivity or climate exposure are measured are unlikely to produce accurate or robust results. Additionally, it's shown how we also need to understand the underlying ecological drivers (in this case habitat and precipitation) if we are to understand how it may change going forward. From what I know of this field, these are significant results; however, this is a field (species' climate sensitivities) that is only tangential to my own research.

On the whole, I found the methodology robust, well explained, and appropriate for the analysis. The analysis supports the conclusions that are drawn.

Although I did not review the original submission, I have taken a look at the response to reviewers and am satisfied that the authors have clarified these. I think the new framing (not focusing so much on the temperature window analysis) highlights the importance and significance of the results.

I have a few minor comments on the manuscript:

L187-188: I had to think this through a couple of times - perhaps highlight that you are referring to the mean across years, rather than the mean within the year (I think it's talking about how the first clutch is defined right before this that confused me).

L233-234: brief description of this metric (probability of a change in AIC_c?)

L245-253: A little more detail is required here as to the structure of the SEMs used. At present I might be able to reproduce this part of the analysis, but I would be making some assumptions about what was done.

L284: reference to Box 1, but this is no longer in the manuscript.

L419: typo - really should be rely.

L464: typo - specie's should be species'

Figure One: This figure is really great and makes it so much simpler to understand the methods and the logic behind them. In description of part a) though it mentions grey shaded area, but it's red/pink.

Reviewer #3 (Remarks to the Author):

Synopsis: This study conducts a phenological analysis of the role of temperature as an important component in the phenology of egg laying dates for two species across their ranges. To do so, the authors seek to explore the nonstationarity of these phenological responses by estimating intraspecific aspects of climate change exposure and phenological sensitivity. I found the MS to be well written and the methods of the climate window analysis to be explained in sufficient detail. The overall findings that phenological sensitivity was mediated by habitat type and that low sensitivity was more likely in areas experiencing the greatest change was compelling. I agree that there are few studies looking at intraspecific patterns of phenological sensitivity across broad geographic scales, however, I had several lingering concerns about certain aspects of the methodology.

Major comments:

I felt like there is a missed opportunity for unpacking the role of why habitat mediates the impact of climate change on phenological responses. There is some nice discussion on how habitat heterogeneity and differences in floral composition might alter the phenology of important food/insect resources and have subsequent effects on the phenology of birds. However, these habitat types also maintain very different microclimates that would undoubtedly influence climate change exposure. There is a rich literature on how forest cover influences the temperature experienced by organisms (Vanwalleghem and Meentemeyer 2009, Suggitt et al. 2011, Frey et al. 2016, Latimer and Zuckerberg 2017, Frenne et al. 2021). I think this could be integrated in your hypotheses, and could potentially point to the importance of landscape context in the study of phenological responses.

Lines 169-179: Given the importance of habitat in this analysis, I was not sure why there was no better integration of a remotely-sensed product that offers global land cover information, such as MODIS. Given that each site has exact coordinates, it seems like using the MODIS Terra+Aqua Combined Land Cover product with 500-m resolution would be better than relying on site descriptions by data owners as to the dominant cover type.

Lines 191-203: The use of climate data from a diversity of sources (gridded products, weather station interpolation) is not ideal. All these data sets have their own limitations and sources of uncertainty. Would it be possible to simply use one data source (e.g., weather station) or compare the temperature data from the gridded products and the weather stations? Perhaps there is some existing accuracy assessment for the gridded dataset? Further, I was left wondering why you would use precipitation data derived from a PCA from another study (line 273-274) instead of the gridded climate or weather station data.

In all analyses, it is important to evaluate and assess multicollinearity among the predictors (Dormann et al. 2013). I am particularly curious/concerned about the correlation between habitat and latitude. Having two highly correlated variables in the same model could cause issues with standard errors and inverted parameter estimates.

Minor comments:

In the first paragraph, you refer to "absolute phenology" and "phenological sensitivity" (Line 85). It is not immediately clear what is the distinction between those two terms. It seems like the absolute phenology is the observed change in a phenological response whereas phenological sensitivity is change in phenology per degree change. It would be useful to avoid additional terms that need additional definition.

Line 89: Why are phenological sensitivity and climate change exposure considered "traits"?

Line 141: Not sure why precipitation is given as an abiotic example instead of temperature.

A minor note, but I felt like there was some repetitiveness of the Arctic amplification pattern throughout the ms. There have been multiple studies describing the warming of northerly latitudes in the northern hemisphere.

Lines 246-253: I was confused how the use of structural equation modeling and non-parametric bootstrapping was able to account for non-climatic factors?

Lines 256-266: I would suggest a more complete description of midpoint, duration, and delay and their interpretation. This section clearly details the predictors, but it is less clear what the response variables are.

Line 398-400: Could this statement be explained a bit better? It is not clear why floral species composition would be less important.

Figure 1. In Box (c), the superscript numbers are switched for climate change exposure (which should be 3) and phenological sensitivity (which should be 4).

Figure 3. There seems to be an important outlier for the evergreen sites. How was this handled?

Figure 4. Were the results for both species grouped in this figure?

References

Dormann, C. F., J. Elith, S. Bacher, C. Buchmann, G. Carl, G. Carr, J. R. Garcia, B. Gruber, B. Lafourcade, P. J. Leit, M. Tamara, C. McClean, P. E. Osborne, B. S. Der, A. K. Skidmore, D. Zurell, and S. Lautenbach. 2013. Collinearity: a review of methods to deal with it and a simulation study evaluating their performance. *Ecography* 36:27–46.

Frenne, P. D., J. Lenoir, M. Luoto, B. R. Scheffers, F. Zellweger, J. Aalto, M. B. Ashcroft, D. M. Christiansen, G. Decocq, K. D. Pauw, S. Govaert, C. Greiser, E. Gril, A. Hampe, T. Jucker, D. H. Klings, I. A. Koelemeijer, J. J. Lembrechts, R. Marrec, C. Meeussen, J. Ogée, V. Tyystjärvi, P. Vangansbeke, K. Hylander, and P. D. Frenne. 2021. Forest microclimates and climate change: Importance, drivers and future research agenda:1–19.

Frey, S. J. K., A. S. Hadley, S. L. Johnson, M. Schulze, J. A. Jones, and M. G. Betts. 2016. Spatial models reveal the microclimatic buffering capacity of old-growth forests. *Science Advances* 2:e1501392–e1501392.

Latimer, C. E., and B. Zuckerberg. 2017. Forest fragmentation alters winter microclimates and microrefugia in human-modified landscapes. *Ecography* 40:158–170.

Suggitt, A. J., P. K. Gillingham, J. K. Hill, B. Huntley, W. E. Kunin, D. B. Roy, and C. D. Thomas. 2011. Habitat microclimates drive fine-scale variation in extreme temperatures. *Oikos* 120:1–8.

Vanwallegem, T., and R. K. Meentemeyer. 2009. Predicting Forest Microclimate in Heterogeneous Landscapes. *Ecosystems* 12:1158–1172.

Reviewer #1 (Remarks to the Author):

This study examines the relationship of date of first egg laid in two species, Great Tit and Blue Tit, and temperature windows across 67 nest-box breeding populations from a wide range of latitude and longitude across Europe. The paper finds that the temperature windows to which these species select laying dates are later at higher latitudes, but no relationship with longitude. Interestingly, the rate of phenological change does not positively scale with changing temperature, rather the opposite. The authors also explore three different broad habitat class, deciduous forest, mixed forest, and evergreen show differing levels of phenological sensitivity, with deciduous forests showing the highest sensitivity (days advanced/degree C). Mixed and evergreen habitats showed comparable rates of phenological sensitivity. The paper is well written and contains few errors throughout. I appreciate the authors taking the time to address comments from a previous review. Figure 1 now nicely describes and illustrates the terms used throughout the paper.

I have no further comments to improve this paper.

Response:

We thank the reviewer for their comments and taking the time to read the manuscript.

Reviewer #2 (Remarks to the Author):

The key results of this paper are to show that there is intra-specific spatial variation in phenological sensitivity and climate exposure; and, in particular that these two measures are negatively correlated in this case. This means that existing methods which use single populations to estimate climate sensitivity across a range, and/or where only one of phenological sensitivity or climate exposure are measured are unlikely to produce accurate or robust results. Additionally, it's shown how we also need to understand the underlying ecological drivers (in this case habitat and precipitation) if we are to understand how it may change going forward. From what I know of this field, these are significant results; however, this is a field (species' climate sensitivities) that is only tangential to my own research.

On the whole, I found the methodology robust, well explained, and appropriate for the analysis. The analysis supports the conclusions that are drawn.

Although I did not review the original submission, I have taken a look at the response to reviewers and am satisfied that the authors have clarified these. I think the new framing (not focusing so much on the temperature window analysis) highlights the importance and significance of the results.

I have a few minor comments on the manuscript:

L187-188: I had to think this through a couple of times - perhaps highlight that you are referring to the mean across years, rather than the mean within the year (I think it's talking about how the first clutch is defined right before this that confused me).

Response: In this sentence we are referring to means *within* a year, such that every population/species combination has a single (average) phenology value in each year. We have now made this more explicit in the text (l. 186-187):

“We used the mean laying date of first clutches within a year as a measure of population phenology for all further analyses.”

By doing this, our analysis does not consider inter-individual variation in phenology within a year that is likely driven by factors other than temperature (e.g. resource availability, phenotypic variation). The uncertainty of these mean estimates varies for different populations and years. We account for this variation in uncertainty by weighting our models with the inverse of the standard error (l. 223-224).

L233-234: brief description of this metric (probability of a change in ΔAIC_c ?)

Response: We have now included a sentence describing the $P_{\Delta AIC_c}$ metric (l. 239-241).

“ $P_{\Delta AIC_c}$ represents the probability that we would observe a given ΔAIC_c value in our temperature window analysis when no relationship exists between temperature and phenology”

L245-253: A little more detail is required here as to the structure of the SEMs used. At present I might be able to reproduce this part of the analysis, but I would be making some assumptions about what was done.

Response: Following comments from both reviewer 2 and 3 we have added more detail for the SEM section of the analysis. We have now included a detailed description of the SEM in the Methods (l. 269-275) and have included a path diagram of the structural equation model used as a supplementary figure (Fig. S1).

“We partitioned the effect of time (year) on laying date into a direct and indirect pathway. The indirect pathway accounted for effects of time that occur via changes in temperature, such as climate change. The direct pathway accounted for any changes in laying date that occurred through effects other than temperature change. The relationship between temperature and laying date in the structural equation model was used as the measure of phenological sensitivity. A path diagram representing the structural equation model is included as Fig. S1.”

L284: reference to Box 1, but this is no longer in the manuscript.

Response: We now refer to Figure 1 at this point in the text.

L419: typo - really should be rely.

Response: Typo fixed.

L464: typo - specie's should be species'

Response: Typo fixed.

Figure One: This figure is really great and makes it so much simpler to understand the methods and the logic behind them. In description of part a) though it mentions grey shaded area, but it's red/pink.

Response: Figure 1 has been updated to fix the colour of the shaded area.

Reviewer #3 (Remarks to the Author):

Synopsis: This study conducts a phenological analysis of the role of temperature as an important component in the phenology of egg laying dates for two species across their ranges. To do so, the authors seek to explore the nonstationarity of these phenological responses by estimating intraspecific aspects of climate change exposure and phenological sensitivity. I found the MS to be well written and the methods of the climate window analysis to be explained in sufficient detail. The overall findings that phenological sensitivity was mediated by habitat type and that low sensitivity was more likely in areas experiencing the greatest change was compelling. I agree that there are few studies looking at intraspecific patterns of phenological sensitivity across broad geographic scales, however, I had several lingering concerns about certain aspects of the methodology.

Major comments:

I felt like there is a missed opportunity for unpacking the role of why habitat mediates the impact of climate change on phenological responses. There is some nice discussion on how habitat heterogeneity and differences in floral composition might alter the phenology of important food/insect resources and have subsequent effects on the phenology of birds. However, these habitat types also maintain very different microclimates that would undoubtedly influence climate change exposure. There is a rich literature on how forest cover influences the temperature experienced by organisms (Vanwalleghem and Meentemeyer 2009, Suggitt et al. 2011, Frey et al. 2016, Latimer and Zuckerberg 2017, Frenne et al. 2021). I think this could be integrated in your hypotheses, and could potentially point to the importance of landscape context in the study of phenological responses.

Response: We agree that a discussion of microclimatic temperature is important for the manuscript, particularly as gridded datasets like the E-OBS Gridded Dataset used here will not account for microclimatic differences between habitats (e.g. Latimer and Zuckerberg 2017). However, it is less clear how we should expect microclimatic differences to affect climate change exposure or phenological sensitivity. Macro- and microclimatic patterns of climate change exposure ($^{\circ}\text{C}/\text{time}$) are expected to be strongly related (Zellweger et al. 2020 <https://doi.org/10.1126/science.aba6880>; see their Figure 1C). Although different habitat types will maintain different microclimates, we believe the change in these microclimates should be well estimated by climate change exposure from our gridded dataset. It is possible that changes in habitat characteristics over time could alter climate change exposure by affecting the extent of microclimatic buffering between years; however, we do not have any longitudinal data on habitat characteristics at our study sites, such as canopy height or cover, that might allow us to investigate this further. Having said this, we still think that including information about microclimates will improve the manuscript, particularly as a failure to account for microclimate (or habitat/landscape scale characteristics) may drive a large amount of the unexplained variance we observe in our phenology data. To address this, we have included a paragraph in the discussion to highlight the limitations of using macroclimatic measurements and the potential importance of measuring other habitat/landscape characteristics in the study of phenology (l. 471 - 485).:

“Quantification of phenological sensitivity and climate change exposure in our analysis relies on temperature data from the European wide E-OBS Gridded Dataset. Gridded datasets are an invaluable tool to study macroclimatic temperature change at a broad spatiotemporal scale, but such datasets do not account for local

microclimatic variation that is more relevant to organismal behaviour. Gridded datasets can differ from local microclimatic temperature by as much as 4°C, mediated by both habitat and landscape scale characteristics, such as canopy cover and degree of habitat fragmentation. Although macro- and microclimatic trends in climate change exposure are strongly related, failure to account for microclimates will likely contribute to additional unexplained variation in phenology, which may be more pronounced in those habitats where macro- and microclimatic temperatures diverge most, such as those near urban areas. Detailed fine-scale temperature measurements were unavailable over the large spatiotemporal scale used in our study, making it impossible to directly incorporate effects of microclimatic variation in our analysis. In the future, accounting for differences in habitat characteristics that are known to influence microclimate may provide a feasible alternative to directly measuring microclimates particularly for variables that can be estimated through remote sensing, such as canopy cover or habitat fragmentation.”

Lines 169-179: Given the importance of habitat in this analysis, I was not sure why there was no better integration of a remotely-sensed product that offers global land cover information, such as MODIS. Given that each site has exact coordinates, it seems like using the MODIS Terra+Aqua Combined Land Cover product with 500-m resolution would be better than relying on site descriptions by data owners as to the dominant cover type.

Response: We thank the reviewer for their useful comment and the remote sensing product suggestion. Following the reviewer’s suggestion and a discussion with a remote sensing expert, we tested the MODIS Terra+Aqua Combined Land Cover product (MCD12Q1) as an alternative method to assign habitat type categories, using the 6 included data sets that distinguished between deciduous, evergreen and mixed forests (IGBP, UMD, LAI, BGC, PFT, LCCS1).

We first classified habitat type based on the majority consensus across all 6 data sets. Unfortunately, using this approach 90% of sites (36/40) were not able to be classified into deciduous, evergreen, or mixed; instead, sites were assigned to categories such as ‘sparse forest’, ‘wetland’, or ‘urban’ that are not useful for our analyses. There were only 2 sites (East Dartmoor, UK and Rome, Italy) where a majority of data sets agreed on a usable habitat classification (deciduous). In two other cases (Estonia and Nagshead, UK) the 6 datasets were evenly split between a deciduous and mixed assignment.

To investigate this further, we investigated the ability for an individual layer to classify sites as deciduous, evergreen, or mixed. Of the 6 datasets, the BGC and PFT data sets most successfully classified sites, but these data sets still only provide a useful classification at a minority of sites (18/40; 45%). The other 4 layers only managed to classify 10% (4/40). We have included plots below to visualise these classification outcomes where all classification that were not ‘deciduous’, ‘evergreen’ or ‘mixed’ are grouped as ‘other’.

The inability of MCD12Q1 to classify our sites into relevant habitat types similar to those classified by data owners likely reflects the fact that classification by data owners is focused on vegetation that is ecologically relevant for great and blue tits (e.g. large trees like oak that are used for nesting/foraging), while the remote sensing product considers all land cover regardless of the ecological relevance to our study species. Because of these limitations, MCD12Q1 cannot provide a stand-alone option for habitat type classification in our analysis.

To avoid using a mixed classification from both remote sensing and data owner classification we have kept our original habitat type classification method.

We agree with the reviewer that the use of remote sensing products is valuable and important in these types of analyses. Although in this case we have found that remote sensing data does not supersede our current approach, it has potential for other purposes. As part of the new discussion of microclimates (discussed above), we highlight the potential for remote sensing tools to quantify additional habitat characteristics, such as canopy cover or habitat fragmentation, that might be used to account for microclimatic variation (l. 482 - 485).

“In the future, accounting for differences in habitat characteristics that are known to influence microclimate may provide a feasible alternative to directly measuring microclimates particularly for variables that can be estimated through remote sensing”

Figure 1: Stacked bar chart showing the habitat type classification of all 40 study sites using 6 available MODIS data sets. In 90% of sites (36 sites) the majority of MODIS data sets classified sites as ‘other’ (i.e. classifications that do not match ‘deciduous’, ‘evergreen’ or ‘mixed’).

Figure 2: Habitat type classification of all 40 study sites using 6 available MODIS data sets. The data sets BGC (BIOME-Biogeochemical Cycles) and PFT (Plant Functional Types) were most successful but still failed to classify sites to ‘deciduous’, ‘evergreen’ or ‘mixed’ in the majority of cases (22 sites; 55%).

Lines 191-203: The use of climate data from a diversity of sources (gridded products, weather station interpolation) is not ideal. All these data sets have their own limitations and sources of uncertainty. Would it be possible to simply use one data source (e.g., weather station) or compare the temperature data from the gridded products and the weather stations? Perhaps there is some existing accuracy assessment for the gridded dataset?

Response: We agree with the reviewer that reliability and consistency of temperature data is vital for our analysis and that we have not properly justified our choice of temperature data. The E-OBS Gridded Dataset that we use for all but two of our sites (Vlieland and Sicily) is built using data from a European wide weather station network. E-OBS use this weather station network to create blended time series that allows them to complete interrupted time series using data from nearby weather and synoptical stations. They also flag and remove those observations that are not considered ‘valid’ (the full documentation for these processes can be found here: <https://knmi-ecad-assets-prd.s3.amazonaws.com/documents/atbd.pdf>). We believe the E-OBS Gridded Dataset represents the best European wide climate dataset available and could not be reliably replaced by analysis of individual weather stations alone as this would fail to incorporate the time series completion and observation validation provided by E-OBS. We have now included a more detailed description of the gridded dataset in the text to make it clearer to the reader what this gridded dataset represents and its robustness (l. 191-197).

“The gridded dataset uses blended weather time series from the European wide weather station network of the European Climate Assessment & Dataset project (ECA&D; <https://www.ecad.eu/>). Blended time series use data from nearby weather and synoptic stations to extend and fill in gaps in existing weather station time series.

The E-OBS Gridded Dataset uses all observations from available blended time series that are considered 'valid' under the ECA&D quality control rules. Full documentation explaining blending and quality control rules can be found on the ECA&D website."

However, we appreciate this does not overcome the problem presented by the two sites where the gridded dataset is not used. To ensure that our results were not affected by the choice of temperature data used at these two sites we also re-fitted our models with these two populations removed, which did not affect our conclusions. This additional test has now been described in the text (l. 208-211) and detailed in supplementary material.

"To ensure that our results were robust to the choice of temperature data in Vlieland and Sicily we conducted supplementary analyses with these populations excluded (Supplementary material 2). Our results were consistent with these populations removed."

Further, I was left wondering why you would use precipitation data derived from a PCA from another study (line 273-274) instead of the gridded climate or weather station data.

Response: Including precipitation data in our analysis raises the same issue of window selection as we describe for temperature windows in l. 111-125; however, fitting climate windows for precipitation can be more difficult due to the inherently high variability of the data and can become much more complex if interactions with temperature are expected. Furthermore, availability of precipitation data in the E-OBS Gridded Dataset is patchier than for temperature, particularly in early years. We considered the PCA from Metzger et al. to be a plausible alternative that does not face these same issues. The PCA from Metzger et al. provides a broad quantification of annual precipitation patterns across Europe using a range of precipitation variables sampled at different points of the year. In this way, the PCA values are more akin to an annual 'precipitation regime' for each site that reflects broad differences in annual precipitation patterns. Although using this PCA does not account for population-specific precipitation windows, it should still provide a robust measure of differences in annual precipitation characteristics between sites. A more thorough description of the PCA and why it was used is provided in the text (l. 290-295).

"We included a fixed effect to account for differences in annual precipitation patterns between populations. We derived a precipitation metric from the principal component analysis of Metzger et al. which incorporates variation in precipitation (mm) across multiple months over the year to provide a quantification of annual precipitation patterns at each study site. A higher PCA value represents a site with higher precipitation."

In all analyses, it is important to evaluate and assess multicollinearity among the predictors (Dormann et al. 2013). I am particularly curious/concerned about the correlation between habitat and latitude. Having two highly correlated variables in the same model could cause issues with standard errors and inverted parameter estimates.

Response: We have now included a more robust check of multi-collinearity in our model and described this in the methods. For all models we calculated generalised variance inflation factor (gVIF) of each predictor variable to assess multi-collinearity, using a cut-off of

3 to signal multi-collinearity. None of our variables exceeded our gVIF cut-off, suggesting no major issues with multi-collinearity (l. 319-322).

“Generalised variance inflation factors were used to test for multi-collinearity between variables in all models with a cut-off value of 3. No variables exceeded our variance inflation factor cut-off, suggesting that multi-collinearity between model predictors was not a concern.”

Minor comments:

In the first paragraph, you refer to “absolute phenology” and “phenological sensitivity” (Line 85). It is not immediately clear what is the distinction between those two terms. It seems like the absolute phenology is the observed change in a phenological response whereas phenological sensitivity is change in phenology per degree change. It would be useful to avoid additional terms that need additional definition.

Response: The reviewer is correct that ‘absolute phenology’ here refers to observed change in phenological response (i.e. laying date). To avoid additional terms we have reworded this sentence (l. 84-85):

“For example, differences in the timing and availability of resources, due to factors such as habitat type, can affect phenology directly and also alter phenological sensitivity”

Line 89: Why are phenological sensitivity and climate change exposure considered “traits”?

Response: ‘traits’ has been changed to ‘variables’

Line 141: Not sure why precipitation is given as an abiotic example instead of temperature.

Response: Phenological sensitivity is already necessarily a function of temperature (i.e. laying date/ $^{\circ}\text{C}$), therefore we focussed on another abiotic variable (precipitation).

A minor note, but I felt like there was some repetitiveness of the Arctic amplification pattern throughout the ms. There have been multiple studies describing the warming of northerly latitudes in the northern hemisphere.

Response: We have removed several statements about Arctic amplification from the text to reduce repetitiveness.

Lines 246-253: I was confused how the use of structural equation modeling and non-parametric bootstrapping was able to account for non-climatic factors?

Response: A lack of detail in the description of the structural equation model was also raised by reviewer 2. To address the concerns of both reviewers we have now included a

more detailed description of the SEM in the Methods (l. 269-275) and have included a path diagram of the structural equation model used in the supplementary material (Fig. S1).

“We partitioned the effect of time (year) on laying date into a direct and indirect pathway. The indirect pathway accounted for effects of time that occur via changes in temperature, such as climate change. The direct pathway accounted for any changes in laying date that occurred through effects other than temperature change. The relationship between temperature and laying date in the structural equation model was used as the measure of phenological sensitivity. A path diagram representing the structural equation model is included as Fig. S1.”

Lines 256-266: I would suggest a more complete description of midpoint, duration, and delay and their interpretation. This section clearly details the predictors, but it is less clear what the response variables are.

Response: We have now included a more complete discussion of midpoint, duration, and delay at this point in the text (l. 248-260).

“In all populations where we identified a true temperature cue, we determined the duration and midpoint of the temperature window and the difference between the midpoint and the mean laying date of the population over the study period (delay). The window midpoint is a useful single metric to describe the position of the temperature window within a calendar year and is expected to be later for more northerly populations and those in evergreen habitats due to later resource peaks. Temperature window delay represents the time between when a cue is detected and a response (i.e., egg laying) is exhibited. Delay between a cue and response may be caused by physical restrictions that limit the speed of response (e.g., gonadal development) or may reflect a delay between cues and relevant future conditions when cues act through multiple trophic levels. Previous studies have assumed that intra-specific variation does not exist in this delay variable, but this has not been explicitly tested. Variation in temperature window duration with latitude has been reported in previous studies, although the direction of such an effect has differed between studies making any expectations around duration unclear.”

Line 398-400: Could this statement be explained a bit better? It is not clear why floral species composition would be less important.

Response: We wanted to emphasise that low phenological sensitivity was not restricted to a single species assemblage (e.g. evergreen oak) but was observed across multiple evergreen habitats. The sentence has now been reworded to make this point clearer (l. 419-422).

“The fact that this pattern was observed in both needle-leafed and broad-leafed evergreen forests suggests that low phenological sensitivity is not restricted to a single floral species assemblage but is a common trait of evergreen habitats.”

The reason why the presence of evergreen trees might be more important than any particular species assemblage is covered in our discussion of food resource peaks (l. 424-

438). In short, we hypothesise that evergreen habitats exhibit an extended resource peak and this reduces the fitness cost of mistiming in breeding birds.

Figure 1. In Box (c), the superscript numbers are switched for climate change exposure (which should be 3) and phenological sensitivity (which should be 4).

Response: Superscript numbers are now changed in Figure 1.

Figure 3. There seems to be an important outlier for the evergreen sites. How was this handled?

Response: We investigated the low midpoint value seen in Figure 3. We have no evidence to exclude this point from the analysis as it appears to be a legitimate measurement. This point comes from our furthest south evergreen site (Sagunto, Spain), and so the low midpoint value may be explained by the low latitude at which this site is found. To be certain that this point is not driving our conclusions we re-estimated the effect of habitat type with this point excluded and found a larger significant difference between deciduous and evergreen habitats (Coefficient and CI: 13.48 [8.77 / 19.54]).

Figure 4. Were the results for both species grouped in this figure?

Response: Because we found no strong effect of species in our analysis we decided to show grouped data for both species in Figure 3b and Figure 4. We have made this clearer in the figure caption:

“Shown are box and violin plots of phenological sensitivity of both great and blue tits in each habitat type.”

References

Dormann, C. F., J. Elith, S. Bacher, C. Buchmann, G. Carl, G. Carr, J. R. Garcia, B. Gruber, B. Lafourcade, P. J. Leit, M. Tamara, C. McClean, P. E. Osborne, B. S. Der, A. K. Skidmore, D. Zurell, and S. Lautenbach. 2013. Collinearity : a review of methods to deal with it and a simulation study evaluating their performance. *Ecography* 36:27–46.

Frenne, P. D., J. Lenoir, M. Luoto, B. R. Scheffers, F. Zellweger, J. Aalto, M. B. Ashcroft, D. M. Christiansen, G. Decocq, K. D. Pauw, S. Govaert, C. Greiser, E. Gril, A. Hampe, T. Jucker, D. H. Klimes, I. A. Koelemeijer, J. J. Lembrechts, R. Marrec, C. Meeussen, J. Ogée, V. Tyystjärvi, P. Vangansbeke, K. Hylander, and P. D. Frenne. 2021. Forest microclimates and climate change : Importance , drivers and future research agenda:1–19.

Frey, S. J. K., A. S. Hadley, S. L. Johnson, M. Schulze, J. A. Jones, and M. G. Betts. 2016. Spatial models reveal the microclimatic buffering capacity of old-growth forests. *Science Advances* 2:e1501392–e1501392.

Latimer, C. E., and B. Zuckerberg. 2017. Forest fragmentation alters winter microclimates

and microrefugia in human-modified landscapes. *Ecography* 40:158–170.

Suggitt, A. J., P. K. Gillingham, J. K. Hill, B. Huntley, W. E. Kunin, D. B. Roy, and C. D. Thomas. 2011. Habitat microclimates drive fine-scale variation in extreme temperatures. *Oikos* 120:1–8.

Vanwallegem, T., and R. K. Meentemeyer. 2009. Predicting Forest Microclimate in Heterogeneous Landscapes. *Ecosystems* 12:1158–1172.

Reviewer comments, further review -

Reviewer #3 (Remarks to the Author):

Synopsis: This study conducts a phenological analysis of the role of temperature as an important component in the phenology of egg laying dates for two species across their ranges and seeks to explore phenological responses by estimating intraspecific aspects of phenological sensitivity. This is second time I have reviewed the manuscript. I believe you have addressed my original comments and I appreciate the work you have done to explore the potential for integrating remote sensing products.

Reviewer #3 (Remarks to the Author):

Synopsis: This study conducts a phenological analysis of the role of temperature as an important component in the phenology of egg laying dates for two species across their ranges and seeks to explore phenological responses by estimating intraspecific aspects of phenological sensitivity. This is second time I have reviewed the manuscript. I believe you have addressed my original comments and I appreciate the work you have done to explore the potential for integrating remote sensing products.

Response:

We thank the reviewer for taking the time to re-read the manuscript and their constructive comments in the last round of revisions.